# A gross margin analysis for Nguni cattle farmers in Limpopo Province, South Africa

**Mapule Valencia Nkadimeng**[1☉], **Godswill Makombe**[2☉], **Obvious Mapiye**[3], **Cletos Mapiye**[3], **Isaac Oluwatayo**[4], **Kennedy Dzama**[3], **Cedric Mojapelo**[5], **Naftali Mollel**[6‡], **Jones Ngambi**[7‡], **Madimetja Human Mautjana**[1]*

**1** University of Limpopo, Polokwane, Limpopo Province, South Africa, **2** Gordon Institute of Business Science, University of Pretoria, Pretoria, Gauteng, South Africa, **3** Department of Animal Sciences, Faculty of AgriSciences, Stellenbosch University, Stellenbosch, Western Cape Province, South Africa, **4** Department of Agricultural Economics and Animal Production, University of Limpopo, Polokwane, Limpopo Province, South Africa, **5** Limpopo Department of Agriculture and Rural Development, Polokwane, Limpopo Province, South Africa, **6** Rural Development and Innovation Hub, University of Limpopo, Polokwane, Limpopo Province, South Africa, **7** Department of Agricultural Economics and Animal Production, University of Limpopo, Polokwane, Limpopo Province, South Africa

☉ These authors contributed equally to this work.
‡ NM and JN also contributed equally to this work.
* hmautjana@gmail.com

**Data Availability Statement:** All relevant data are within the paper and its Supporting Information files.

## Abstract

Factors such as increases in population, urbanization, growth in per capita income and changes in consumer taste and preferences are causing gradual increases in livestock product consumption and demand. South Africa is addressing this predicted increase in livestock products demand by commercializing smallholder livestock producers. The Limpopo Industrial Development Corporation (IDC) Nguni Cattle Development Project is an example of such effort. The economic performance of these efforts needs to be evaluated. We use gross margin analysis to evaluate the performance of the Limpopo IDC Nguni Cattle Development Project. Additionally, we use regression analysis to identify factors influencing gross margins. Our results indicate that although smallholders show potential to commercialize, they lack commercial farming experience and require that a strong extension support system be used as one of the strategies to improve profitability. We also noted that individual farmers were more profitable than group farmers. Multiple regression analysis shows that three variables could be used to stimulate gross margin among the Limpopo IDC Nguni Cattle Development Project farmers. These are herd size, distance to market and farm size. Since farm size is a given, policy should focus on assisting farmers to build their herds and to have better access to markets.

## Introduction

Livestock product consumption and overall demand is gradually increasing due factors such as increases in population, urbanization, growth in per capita income and changes in consumer taste and preferences [1, 2]. According to [3], the demand for beef specifically in South

**Funding:** The authors received no funding for this work.

**Competing interests:** There are no competing interests.

Africa is projected to grow by more than 20% over the next ten years or so. In responding to this, various livestock production strategies were initiated in South Africa [4, 5]. Current empirical research shows that most of these initiatives are seemingly confined to increasing animal numbers and environmental protection. However, strategies and policies aiming at monitoring and assessing the profit efficiency and financial competitiveness, especially among the subsistence and emerging smallholder farmers, are limited [3, 6].

Given its geography, the importance of livestock in South Africa can be considered from many perspectives. Invariably, it stands to be the most viable agricultural activity in a large part of the country. Livestock farming has remained spatially important and a multifunctional livelihood strategy for majority of the rural poor [1, 7]. For many years, the sector is ranked high as the main contributor to national agricultural output, with beef sector being the largest farming activity [1, 4]. According to [8], in the 2017/18 production season, the country's agricultural gross farming income based on all agricultural products, recorded a 2,1% increase. This growth was primarily influenced by the increase in income derived from livestock products (13,3%) and horticultural produce (1,8%). Of the total gross value of agricultural production also in that season, livestock contributed approximately 50% with horticultural products and field crops contributing (28%) and (22%) respectively [8]. Apart from the contribution towards agricultural output, livestock are vital in alleviating poverty and driving socio-economic development [7]. Among the rural and resource poor people, livestock contribute substantially to livelihoods; they are a store of wealth, source of income and food (meat and milk), draught power and manure [9, 10] and act as collateral for credit and essential security nets against unforeseen circumstances [11]. Cattle are an important resource to South Africa's livestock sector and the rural poor [7]. Of the estimated 12.55 million cattle available in South Africa, about 5.6 million belong to nearly three million subsistence and emerging smallholder farmers across the country [1]. Thus, the smallholder sector representing 80% of total cattle farmers are a strategic sector in achieving food security. It's therefore, essential to bring them into the formal economy [5].

The South African livestock industry has remained a distinctly dual economy [4]. On one end, it has well-established and highly sophisticated large commercial farmers and on the other end, a struggling smallholder farming sector (subsistence and emerging) [1, 5]. Emerging farmers are beneficiaries of the government's Land Redistribution for Agricultural Development (LRAD) programme [4, 12] which is a strategy adopted by the government to redress the historic land imbalance and bridge the gap of dualism. The LRAD programme largely outmoded the Settlement Land Acquisition Grant (SLAG) model which targeted the resettlement of poor and vulnerable South Africans. Conversely, LRAD model recognizes and assists better resourced and skilled previously disadvantaged farmers (mainly black) in acquiring already existing farms as a step to becoming commercial farmers [12]. This initiative gave birth to the emerging/commercially oriented cattle farmers who are currently transitioning from small commercial scale to large commercial scale farming [13]. Various smallholder farmer empowerment projects have been initiated throughout the country to support land reform including those funded by IDC such as the Limpopo IDC Nguni Cattle Development Project.

## Overview of the Limpopo IDC Nguni Cattle Development Project

The project is a development orientated partnership between IDC, Limpopo Department of Agriculture (LDA) and the University of Limpopo (UL). Implemented as the IDC-Limpopo Nguni Development Trust, the project was founded in 2006. The trust seeks to facilitate the reintroduction of the indigenous Nguni cattle genetic resource into the communal farming areas of Limpopo Province. Specifically, the project was launched to promote the

commercialization of previously disadvantaged rural South Africans through the creation of commercial opportunities in beef cattle farming [14]. Based on the provincial selection criteria, prospective beneficiaries should be residence of the province, owning or having provable access to at least 600ha of land and exhibiting excellent cattle management and general entrepreneurships skills. The project initially recruited 62 beneficiaries who are farming as individuals, cooperatives or as Community Property Associations (CPAs). CPA refers to a group of people in the community having formalized rights to use land [15]. These groups of people benefit from an Act that was established by the Department of Rural Development and Land Reform (DRDLR). The CPA Act No. 28 of 1996 is the main instrument employed by DRDLR to provide land to the groups. Eleven of the Nguni project farmers in the study area were either CPAs or cooperatives. Under the CPA and cooperative arrangements, decisions are made by a group and not by an individual.

Each farm received a loan package valued at R330 000 (Approximately USD48 000 in that period) which consisted of 30 pregnant Nguni heifers/cows and a breeding bull [16]. The equivalent number and type of cattle will be repaid by the farmers to the Trust after a period of five years. A cash payment equivalent to the value of the similar herd as at date of repayment is also acceptable. The repayment package is then used to benefit a newly recruited beneficiary ("Passing of the Gift") hence the project continues to self-scale. Currently, the project is producing high-quality beef meat and hides under free-ranging conditions and has created a sustainable venture in the breeding and conservation of the indigenous Nguni cattle [16].

Overall, majority of the emerging cattle farmers are confronted by various challenges and constraints [4, 13]. The challenges are complex and almost similar to those of the smallholder sector in general [17]. For examples, based on [8], currently, cattle off-take in the smallholder sector is far much lower (9%) relative to the commercial sector (30%). Furthermore, the variations in terms of economic performance and scope is not well captured in the present categorisation of farmers, especially the smallholders [3]. It is of paramount importance to understand the performance of such projects. Therefore, the objective of this study is to analyse the performance of the Limpopo IDC Nguni Cattle Development Project using gross margin as a performance measure. A multiple regression model will also be used to determine the factors that explain variation in gross margins.

## Literature review

The Nguni cattle development Project farmers raise cattle on pasture. Raising cattle on pasture can be taken as an advantage for increasing the gross margins of farmers. Some developed country markets express a very high willingness to pay a premium for pasture raised beef [18: 1]. note that, "Pasture-raised livestock products represent a premium niche with an extra value through a cleaner environmental footprint and care for animal welfare, including wildlife......There are a variety of consumer groups willing to pay a premium for a pasture-raised attribute even on top of an organic price premium." Commercial farmers in South Africa can achieve the pasture attributes by effecting changes in management. However, smallholder farmers' production systems are already inherently low input [19], thus certification for organic and/or pasture production does not need a lot of management adjustments [20]. also studied the willingness of traders and consumers to support the development of a natural pasture-fed beef brand (NPB) by smallholder cattle producers in South Africa. [20: 207] concluded that, "Overall, beef traders and consumers held positive impressions regarding the development of a NPB brand by smallholder cattle producers but were not willing to support its development." This gives an indication of potential which could be exploited in the cattle development strategy in South Africa, however, the farmers like those belonging to the

Limpopo IDC Nguni Cattle Development Project do not have the know-how to enable them-selves to access such markets and significantly increase their gross margins so the South Afri-can government should make strategic action(s) regarding this potential. Due to the sustainable horizontal growth of the Limpopo Nguni Cattle Development Project, it is possible that with time, the farmers can achieve the consistency of the volumes required in the export markets, especially if they coordinate their marketing activities with similar projects around South Africa. The Farm Assured Namibian Meat (FAN Meat) scheme which sells free-range, hormone-free beef with guaranteed animal welfare and veterinary standards, which is described by [21] as a good example of such possibility and could be emulated in South Africa thus positively impacting farmer gross margins. Alternatively, farmers could vertically inte-grate in the livestock value chain by establishing feedlots and potentially slaughterhouses and butcheries [22, 23] showed that feedlots increased farm gross margins for smallholder framers in Burkina Faso.

Local infrastructure also affects farmer gross margins [24]. showed that cattle traders who sold cattle to a terminal market obtained more profit than those who bought for resale to another market. They also showed that a very large proportion of marketing costs was consti-tuted by transport. This may be correlated with poor infrastructure for accessing markets, like poor roads. When traders' profit margins are squeezed by high transport costs, they tend to pass the costs onto producers by over negotiating prices resulting in low gross margins for farmers. Due to lack of market information [24] farmers are usually price takers. Thus, even if farmers choose to sell their animals at farm gate without transporting them, better transport in the areas accessed by traders has a positive impact on framers' gross margins.

The Limpopo IDC Nguni Cattle Development Project farmers can also improve their gross margins through crossbreeding Nguni with exotic breeds. The Limpopo IDC Nguni Cattle Development Project farmers did try cross breeding with Angus bulls [25]. reports that the Limpopo IDC Nguni Cattle Development Project farmers, through the Nguni Development Trust negotiated with a South African Supermarket chain, Pick n Pay so that they could sell the first filial generation of a Nguni X Angus crossbreed and sell it as Angus beef. Angus beef is a branded in South Africa and sells at a high price. This would have a significant impact on increasing the farmers' gross margin. However, as [25] reports, two things happened, firstly, the farmers were not able to satisfy the volumes needed by Pick n Pay in a timely manner, and secondly, the Angus bulls could not stand the high Limpopo temperatures and they started to die. The programme was therefore discontinued. However [26], mention that the gross margin and the benefit-cost ratio were highest for crossbreds in their study in Mali. The study by [26] and the experience of the Limpopo IDC Nguni Cattle Development Project farmers with the Nguni X Angus cross breed demonstrates the fact that the selection of animals being used for a cross breeding programme, in this case the bulls, needs careful consideration for the success of the breeding programme. The latter also demonstrates how the Nguni cattle are adapted to the Limpopo environment where the Angus bulls could not survive.

Gross margins are also affected by the seasonal variation in prices caused by variations in supply and demand which are largely a function of social events like holidays. Availability of feed also affects prices because the Limpopo IDC Nguni Cattle Development Project farmers raise their cattle on rainfed pasture, the condition of the animals is poor during the dry season when they fetch to lowest prices during the year. In Limpopo, drought also drastically reduces gross margins [27]. mention that livestock production is risky during years of drought when it can deliver negative gross margins. Without external earnings and/or accumulated saving, during a drought, cattle farmers can be driven out of business due to negative gross margins [27] because it is not easy to introduce agricultural risk management strategies in developing countries [28]. The Limpopo IDC Nguni Cattle Development Project farmers have neither

external earnings nor accumulated savings which makes a strong case that government has to provide support for them so that the South African livestock industry can achieve one of the national objectives of bringing smallholders who benefited from the land reform programme, like some of the Limpopo IDC Nguni Cattle Development Project farmers, into the mainstream economy.

## Methodology

Gross margin analysis has been used for a long time. Although there are far more advanced techniques that are applied today, gross margin analysis has been and is still used for farm budgeting in farm planning and management [29]. According to [30] gross margin analysis is a simple model that is used to estimate the financial returns to a production process. It is used as a simple proxy for the profitability of a production process. Because of its simplicity, and the lack of data to compute profits, especially under smallholder farm conditions, it is widely applied as part of the evaluation of the economic performance of smallholder agricultural production systems [30, 31]. Gross margin analysis is used to compare returns from different cropping patterns. For instance [32], used gross margin analysis to compare returns from maize based intercropping systems in Nigeria. The results from such comparisons are usually used as input in farm planning. It can also be used for analyzing cropping practices and cost structures for the same crop. For instance [33], used gross margin analysis to compare cropping practices and cost structures of the production of conventional table potatoes for six countries in Europe, namely; Czech Republic, Denmark, Italy, Poland, Portugal and Slovakia. We think because of the ease of computing both the revenue and costs in the production system, gross margin analysis has largely been applied to cropping enterprises. There are many examples of the application of gross margins analysis to cropping enterprises [32]. applied it to compare returns from maize-based intercropping systems [33], used it for potato production [31], applied it to mungbean production [34], applied it to cassava production [30], applied it for sesame production while [35] used gross margin analysis to assess the impact and financial viability of rural women's food security projects. In most of these studies gross margin was used as a proxy for profit and therefore to estimate the returns from the various enterprises. Gross margin can be used to analyse different types of livestock, for instance [36], used gross margin analysis to evaluate returns from fish production, it appears that there are fewer studies that applied gross margin analysis to livestock production as compared to crop production. In this study we apply gross margin analysis to cattle production.

## Computation of gross margin

Input-output data from the different Nguni smallholder farmers were used to compute gross margins. Gross margin is the value of the output of an individual enterprise (gross value of production), less the variable costs directly attributable to generating the value [30, 37] Gross margin does not take account of fixed costs.

The gross margin relationship is stated as follows:

$$\textbf{Gross margin} = (\textbf{\textit{TR}} - \textbf{\textit{TVC}}) \tag{1}$$

Where: 1. TR = Total Revenue (From livestock sales)

2. TVC = Total Variable Costs. These included the following costs, feeds, labour cost, fuel, transport costs (to market and to bring in inputs), electricity, maintenance (such as costs of maintaining the feeding pens), and animal health costs (such as the costs of medicine). Those costs that needed to be proportionately apportioned to the specific animals that were sold were

proportionately computed. The data necessary for the computation of the gross margin were collected and calculated from individual farmers and CPAs.

## The multiple regression model for gross margin

Regression analysis is rarely used to analyse factors that influence gross margin. Usually models are used for production and yield [36]. used multiple regression to explain the production of fish. In this study we use multiple regression analysis to analyse the factors that influenced the gross margin for the Nguni IDC Project farmers. The model was specified as follows:

$$\mathbf{Y_i = \beta_0 + \beta_1 X_1 + \beta_2 X_2 + \beta_3 X_3 + \beta_4 X_4 + \beta_5 X_5 + \beta_6 X_6 + \beta_7 X_7 + U} \tag{2}$$

Where:
$Y_i$: is the gross margin
$B_0$: constant of the equation;
$B_i$: coefficients of explanatory variables; (where i = 1 to 7)
$X_i$: independent or explanatory variables; (where i = to 7)
U: error term or unexplained variation;
When the explanatory variables are included in the model specification it reads as follows:

$$\mathbf{Li = \beta_0 + \beta_1 \text{ education} + \beta_2 \text{ farm experience} + \beta_3 \text{ extension service} + \beta_4 \text{ distance}}$$
$$\mathbf{+ \beta_5 \text{ herd size} + \beta_6 \text{ farm size} + \beta_7 \text{ marketing agency} + U_t} \tag{3}$$

Ensuing are the variable units of measurement, explanations/justifications and the expected signs.

1. Education (years): The education level of the farmer in years. Farmers who are educated are expected to interpret extension messages better so a positive relationship is expected with gross margin.

2. Farm experience (years): Experienced farmers are expected to have higher gross margins better so a positive relationship is expected with gross margin.

3. Extension service (1 if farmer has access to extension service, otherwise 0). Access to extension services is expected to improve productivity so a positive relationship is expected with gross margin.

4. Distance (km): *Ceteris paribus* there should be a negative relationship between distance to market and gross margin. However, it is possible that better markets can be far away so this sign cannot be predicted *apriori*.

5. Herd size (number): As herd size increases offtake is expected to increase so a positive relationship is expected with gross margin.

6. Farm size (ha): farmers with larger farms generally have more access to more grazing and therefore a positive relationship is expected with gross margin.

7. Marketing agency (1 if a farmer belongs to a marketing agency, 0 otherwise). Farmers who belong to a marketing agency are expected to achieve higher gross margins because of realizing better prices therefore a positive relationship is expected with gross margin.

## Data collection

Data for estimating gross margin were collected from the sample of farmers using structured questionnaires. One farmer was attending a funeral during the time when data were collected so, data were collected from 61 farms. Data were collected on sales from each farm and the prices. The variable costs of maintaining the herd, such as labour, feed and medicines administered to the herd, were estimated for the herd and then proportionately apportioned to the animals that were sold. If animals were pen fattened before sale, the pen fattening costs were also estimated. Income from by-products like offals, hides and horns was also estimated. Data were collected for a period of one year (2015). Ethical requirements, as guided and approved by the University of Limpopo Ethical Committee, were followed. Participants signed a consent form, were informed that they could stop the interviews at any time, that there were no consequences for non-participation and that data would be treated confidentially.

## Results and discussion

### Gross margin estimates

Table 1 summarises the gross margin estimates. The gross margins were calculated from 61 smallholder livestock farmers who provided required data. Data were collected from 11 group managed farms (CPAs or cooperatives) and 50 individually managed farms. The average gross margin for group managed farms was R6 031 while for individual farmers was R15 281. The fact that the individual farmer gross margin is higher than that of the group managed farms can be explained by the fact that, in the group managed farms the decision making, for example, the production and sale of livestock may be complex, requiring more consultation than that of an individual farm. Three of the group managed and 25 of individually managed farms made losses.

### Levels of gross margin model variables

Table 2 summarises the levels of the model variable [38]. argues that education is key to understanding basic principles of farming because farmers with education have an understanding of production techniques, marketing and existing opportunities. The minimum years of education was one and the maximum was 22. Table 2 shows that the average years of education were 13. This means some farmers in the Nguni Project had tertiary qualifications. This might assist them in understanding the acquisition of farming knowledge than the average smallholder farmer.

In terms of farm experience, the minimum farming experience of the smallholder was three years and the maximum was 34 years. The average farming experience was 12 years. It must be mentioned that these farmers are being assisted to transition from smallholder to commercial

**Table 1. Gross margin estimates (n = 58).**

| Farm type | Item | Gross value (ZAR) | TVC (ZAR) | Gross margin (ZAR) |
|---|---|---|---|---|
| Group managed (n = 11) | Total | 1 047 002 | 980 661 | 66 341 |
| | Average | 95 182 | 89 151 | 6 031 |
| Individual (n = 50) | Total | 5 460 599 | 4 696 566 | 764 033 |
| | Average | 109 212 | 9 3931 | 15 281 |

Source: Nkadimeng[37] (2019) with updated data.

**Table 2. Levels of model variables (n = 50).**

| Variable | Value |
|---|---|
| Education (years) | '13(4) |
| Farm experience (years) | 12(9) |
| Extension service (% with access) | 82 |
| Distance to market (km) | 37(36) |
| Herd size (Number) | 134(111) |
| Farm size (ha) | 1281(953) |
| Marketing agency (% participating in marketing agency | 12 |

Source: [25] with updated data.

farming, so most of their experience is from smallholder farming. Its relevance in the commercial farming sector is questionable.

In terms of extension, over 37% of the farmers had contact with a government extension officer once a month (Fig 1). More than 16% of the respondents had contact with a government extension officer once in three months, while 11% of the respondents had contact with a government extension officer once in four months. Sixteen percent of the respondents never had contact with a government extension officer, whereas 5% of the respondents stated they only have contact with a government extension officer on request. Fig 1 shows a schematic representation of extension contact by the farmers.

This level of variation in the extension contact by farmers raises questions about its consistency and effectiveness. The average distance to market was 37 km. The maximum distance

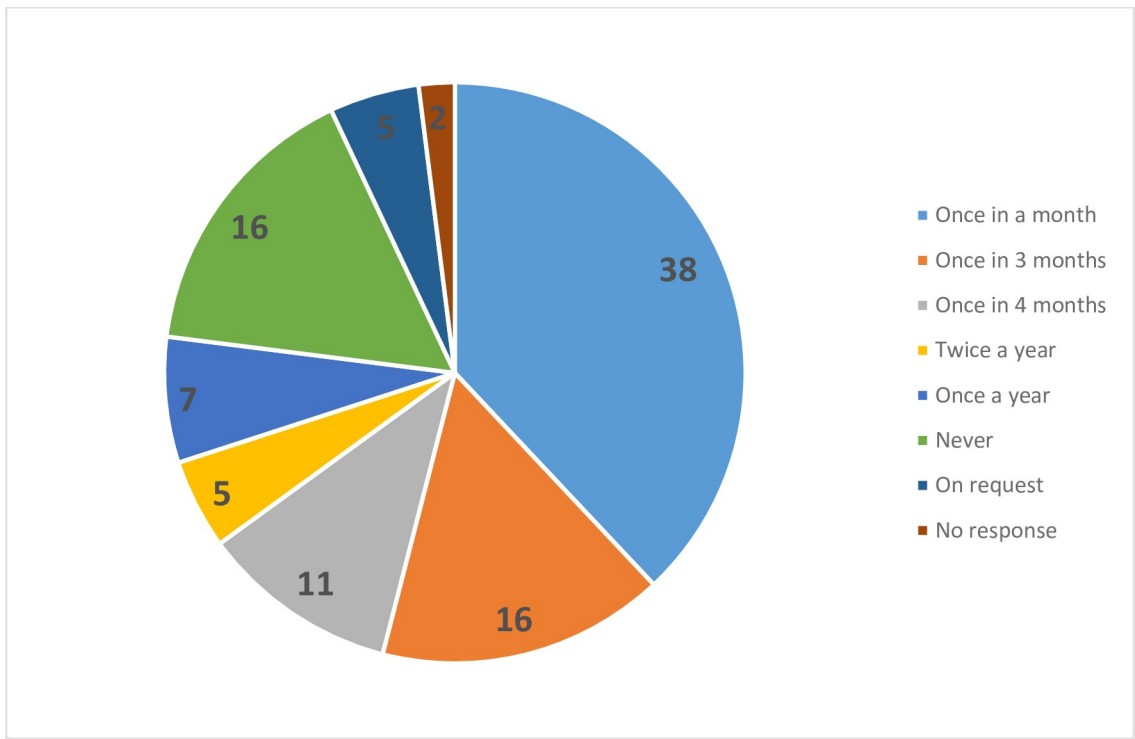

**Fig 1. Farmer extension contact.** Source: [25].

travelled to market was 150 km, while minimum distance travelled to market was one (1) km. This shows that, even though there are some markets which are near, farmers are sometimes willing to travel long distances, perhaps for better prices. The minimum number of cattle owned by a farmer was 36, and the maximum was 600. The average herd size was 134 cattle with standard deviation of 111. The smallest farm among the respondents was 85 ha and the maximum was 4 600 ha with an average of 1 281 hectares. These farms were acquired through the government land reform programme. The farm sizes show the desire of the government to transform the farmers from smallholders to commercial farmers by changing those variables within their means [39]. found that if farmers have a marketing agency, this positively influences profitability. Only 12 percent of the farmers participated in a marketing agency.

## Model estimation results

Table 3 summarises the results of the estimation of the gross margin model. The adjusted $R^2$ was 0.47. In regression analyses in the social sciences where data is sometimes based on farmer recall, an $R^2$ in this range is quite typical [36]. found an $R^2$ of 0.46 in explaining the variation in fish production. The F-test statistic was 7.22 and significant at 0.01. This means that the model explained 47% of the variation in gross margin.

Three out of the seven hypothesized variables, namely herd size, distance to market and farm size were significant. The coefficient of herd size was significant at 1% with the expected positive sign. This implies that as the herd size increases the gross margin also increases. This finding is supported by [40] who argues that large herds generate a higher marketable surplus than small herds. Therefore, even though loan repayment may initially reduce market participation, farmers should focus on herd building after completing loan repayment. The coefficient of farm size was significant with the expected positive sign. This implies that an increase in the size of the farm leads to an increase in the gross margin. This finding is consistent with the results of [39]. Even though farmers do not have the ability to increase their farm size, this finding supports the argument that farmers should take the initiatives to improve factors like the quality of their grazing as this may improve farm productivity in a similar manner to farm expansion. The coefficient of the distance to market was significant with an unexpected positive sign. Based on the argument that increasing distance to market increases the costs that

**Table 3. Gross margin model estimation results (n = 50).**

| Variable | Coefficient | Standard error | t-ratio |
|---|---|---|---|
| Education (years) | -1847.358 | 3154.742 | -0.59 |
| Farm experience (years) | -754.091 | -1390.899 | -0.54 |
| Extension service (% with access) | 12892.050 | 30207.38 | 0.43 |
| Distance to market (km) | 833.296* | 355.454* | 2.34* |
| Herd size (Number) | 439.299** | 117.776** | 3.73** |
| Farm size (ha) | 40.817* | 13.430* | 3.04* |
| Marketing agency (% participating in marketing agency | -26082.710 | -37200.350 | −0.70 |
| Constant | -101587.900 | -53960.24 | -1.88 |
| **Model summary** | | | |
| Adjusted R Square | 47 | | |
| F-Test | 7.22** | | |

* = Significant at 0.05

** = Significant at 0.01.

Source: [25] with updated data.

vary this result may appear counter-intuitive, however, it is possible that the more distant markets are the ones from which the farmers realised higher prices. This relationship certainly warrants further investigation as it is also inconsistent with the finding of [39, 41], who found the opposite relationship.

Though not significant, the negative signs on education and farm experience deserve commenting on. Generally, this sample of farmers was more educated than the average of smallholder farmers. This might mean that the more educated farmers had other income generating sources than the Limpopo IDC Nguni Development Project. In terms of farm experience, as mentioned earlier, most of the sample farmers do not have commercial farm experience. They have smallholder experience and the value of smallholder experience to commercial farming may be questionable.

Another aspect worth noting is that labour should have been included in the gross margin model. However, labour turned out to be correlated to herd size and the model was better with herd size than with labour. Furthermore, marketing agency was included even though there was very little variation in this variable. When the variable was excluded, the signs on herd size and distance to market reversed. We think this is evidence that this is a very strong variable in explaining gross margin such that even though not significant, this is one aspect that should be developed by the Nguni Development Trust in order to improve the commercialisation chances of the Limpopo IDC Nguni Development Project farmers.

## Conclusion

Slightly under a third of the group and slightly under half of the individually managed farms made losses. The government should make an effort to address this because the Limpopo IDC Nguni Cattle Development Project farmers do not have commercial farming experience. Strong extension support can be one of the strategies that can be used to improve the profitability of the farmers [13]. observes that the farmers could benefit from an extension support system based on the currently available electronic technologies like phone accessible applications [42]. observe that the Limpopo IDC Nguni Cattle Development Project farmers could benefit by better information sharing among themselves and the information sharing fostered through extension. Electronic technologies can go a long way in achieving this. This is an area where the government can aggressively support the farmers.

Individually managed farms achieve gross margins that are 2.7 times more than those of group managed operations. Although this aspect requires further study, we conjecture this is a function of the decision-making processes that are more complex for group managed operations. Group members may need training in managing group operations so as to be able to make more efficient decisions. This is particularly true for the Limpopo IDC Nguni Cattle Development Project because, before being part of these commercially oriented groups, they were involved in individual smallholder operations.

The multiple regression analysis shows that there are three variables that could be used to stimulate gross margin among the Limpopo IDC Nguni Cattle Development Project farmers. These are herd size, distance to market and farm size. Farmers need to be encouraged to build their herd sizes, especially after completing loan repayment. There is empirical evidence showing that offtakes, and gross margins are higher for larger herds. The results on the distance to market suggests that farmers may need to be assisted in terms of market access. Gross margins increase as farmers' access more distant market. By assisting farmers to access more distant markets, possibly because they realize better prices, this relationship can be strengthened. Alternatively, local markets can also be developed where farmers could realise prices similar to those of the currently accessed distant markets. The results show that as farm size increases

gross margin increases. Farm sizes for these farmers are finite. Therefore, the farmers need technologies that enable them to more efficiently use the farms they have. This might include fortified grazing which can make farmers realise the gains that are similar to farm expansion, including stronger, more efficient extension support. Participation in marketing agency was not significant. However, given that we could not omit this variable even given its low variation, suggests that it is a strong variable in explaining gross margin variation. We therefore urge the encouragement of the Limpopo IDC Nguni Cattle Development Project farmers to participate in marketing agencies. We also posit that there is a link between this variable and the distance to market because participation in marketing agency can lead to the development of local markets, better access of distant markets or both.

## Limitations

One limitation of this study is that computation of gross margin, although farmers sold at different times of the year, the study did not adequately capture seasonal variation in prices. Seasonal variation in prices is caused by variations supply and demand which are a function of social events like holidays. Availability of feed also affects prices because the Limpopo IDC Nguni Cattle Development Project farmers raise their cattle on rainfed pasture, the condition of the animals is poor during the dry season when they fetch to lowest prices during the year.

## Supporting information

**S1 Data. Gross margin equation data.**
(XLSX)

## Author Contributions

**Conceptualization:** Mapule Valencia Nkadimeng, Godswill Makombe, Cletos Mapiye, Isaac Oluwatayo, Kennedy Dzama.

**Data curation:** Mapule Valencia Nkadimeng, Godswill Makombe, Obvious Mapiye, Cedric Mojapelo.

**Formal analysis:** Mapule Valencia Nkadimeng, Godswill Makombe, Madimetja Human Mautjana.

**Investigation:** Obvious Mapiye.

**Methodology:** Mapule Valencia Nkadimeng, Godswill Makombe, Isaac Oluwatayo, Kennedy Dzama, Naftali Mollel, Jones Ngambi.

**Resources:** Cletos Mapiye, Isaac Oluwatayo, Kennedy Dzama, Cedric Mojapelo.

**Supervision:** Godswill Makombe, Cletos Mapiye, Isaac Oluwatayo, Kennedy Dzama.

**Validation:** Cedric Mojapelo, Naftali Mollel, Jones Ngambi.

**Visualization:** Mapule Valencia Nkadimeng, Godswill Makombe, Cletos Mapiye, Isaac Oluwatayo, Kennedy Dzama.

**Writing – original draft:** Mapule Valencia Nkadimeng, Godswill Makombe, Obvious Mapiye, Isaac Oluwatayo.

**Writing – review & editing:** Mapule Valencia Nkadimeng, Obvious Mapiye, Isaac Oluwatayo.

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
