## [Decision Letter · Decision Letter 0]

19 Mar 2021

PONE-D-21-03815

A gross margin analysis for Nguni cattle farmers in Limpopo Province, South Africa

PLOS ONE

Dear Dr. Madimetja Human Mautjana,

Thank you for submitting your manuscript to PLOS ONE. After careful consideration, we feel that it has merit but does not fully meet PLOS ONE’s publication criteria as it currently stands. Therefore, we invite you to submit a revised version of the manuscript that addresses the points raised during the review process.

We look forward to receiving your revised manuscript.

Kind regards,

László VASA, PhD

Academic Editor

PLOS ONE

Journal Requirements:

Reviewers' comments:

Reviewer's Responses to Questions

**Comments to the Author**

1. Is the manuscript technically sound, and do the data support the conclusions?

Reviewer #1: Partly

Reviewer #2: Yes

2. Has the statistical analysis been performed appropriately and rigorously? 

Reviewer #1: Yes

Reviewer #2: Yes

3. Have the authors made all data underlying the findings in their manuscript fully available?

Reviewer #1: Yes

Reviewer #2: Yes

4. Is the manuscript presented in an intelligible fashion and written in standard English?

Reviewer #1: No

Reviewer #2: Yes

5. Review Comments to the Author

Reviewer #1: The paper deals with gros margin research which wouldn't be enough for a scientific paper but the do it in an empirical way and using multiple regression modell for evaluation.

There is no literature review in the article; it should be written in 2-3 pages, in a critical analytical and co prehensive way.

The content of the chapter "Objectives" is rather belonging to the introduction, no need for a chapter woth one sentence...

Methodology is described well

Sample size isn't too big, but actually enough, taking the size of the geographical area and the affected population into consideration.

There are no limitations described.

In the conclusions, I couldn't find what is the relevance of this research, what it can add to the existing knowledge and what are the consequences for the policy making and practice.

There are too many subchapters in the text, it should be restructured. In its current form, it is npt eligible for the journal's scientific writing criteria.

Reviewer #2: The manuscript deals with a timely question. The analysis of the cattle farming in South Africa is a topic that is interesting for not only domestic (South-African) but international researchers as well. The analysis completed in the manuscript is able to give a thorough overview on the investigated topic and to draw conclusions valid overall. On the other hand, the number of cited literature is not that high in the present form of the publication. I recommend to strengthen the literature review part of the manuscript with citing more sources. With this addition the Authors can widen the soundness of the research completed.

6. PLOS authors have the option to publish the peer review history of their article (what does this mean?). If published, this will include your full peer review and any attached files.

Reviewer #1: No

Reviewer #2: No

---

## [Author Response · Author response to Decision Letter 0]

23 May 2021

Data availability on request

The data for this manuscript is available on request. The university of Limpopo requires that if data is to be shared from student’s thesis, such as is the case with our article, then a request can be sent to the corresponding author upon receipt of which the corresponding author approaches that University of Limpopo’s Research Ethical Committee which ensures that the data are sufficiently de-identified after which the corresponding author can submit data to the requestor.

Reviewer comment Authors’ response

Reviewer #1 

There is no literature review in the article; it should be written in 2-3 pages Literature added.

The content of the chapter "Objectives" is rather belonging to the introduction The chapter “Objectives” removed. Combined with preceding chapter 

There are no limitations described. Limitations added

In the conclusions, I couldn't find what is the relevance of this research, what it can add to the existing knowledge and what are the consequences for the policy making and practice.

 Conclusion revised for policy relevance.

There are too many subchapters in the text One chapter removed. However, we feel that the remaining chapters are necessary for better understanding.

Reviewer #2. 

On the other hand, the number of cited literature is not that high in the present form Literature added.

---

## [Editor Report · Decision Letter 1]

10 Jun 2021

A gross margin analysis for Nguni cattle farmers in Limpopo Province, South Africa

PONE-D-21-03815R1

Dear Dr. Madimetja Human Mautjana,

We’re pleased to inform you that your manuscript has been judged scientifically suitable for publication and will be formally accepted for publication once it meets all outstanding technical requirements.

Kind regards,

Prof. László Vasa, PhD

Academic Editor

PLOS ONE
---

## [Editor Report · Acceptance letter]

15 Jun 2021

PONE-D-21-03815R1 

A gross margin analysis for Nguni cattle farmers in Limpopo Province, South Africa 

Dear Dr. Mautjana:

I'm pleased to inform you that your manuscript has been deemed suitable for publication in PLOS ONE. Congratulations! Your manuscript is now with our production department. 

Kind regards, 

on behalf of

Prof. Dr. László Vasa 

Academic Editor

PLOS ONE